# Spectral Highways: Injecting Homophily into Heterophilic Graphs

## Abstract

The application of convolution in graphs is at the core of Graph Neural Network (GNN) algorithms that led to the emergence of Graph Representation Learning (GRL). Various algorithms have been proposed over the last few years to solve the classical GRL task of node classification in a transductive setting. It is widely assumed that standard GNNs perform better on graphs with high homophily, i.e., nodes belonging to the same class are highly connected to each other. This assumption has led to the designing of specialised algorithms in the last few years for datasets that do not contain the property of homophily, i.e., heterophilic datasets. In this work, we both challenge and leverage this assumption. We argue that it is not necessary to follow the common trend of designing new algorithms but instead focus on understanding and enriching the data. We present a new technique from the perspective of data engineering that enables better performance on heterophilic datasets by both heterophilic GNN algorithms and non-heterophilic GNN algorithms. Our proposed technique, *Spectral Highways*, enables better connectivity and information flow between nodes in a heterophilic graph. We also draw an analogy between the performance of Spectral Highways and a recently proposed network property, i.e., Adjusted homophily. We conduct experiments on 11 baselines and 8 heterophilic datasets and achieve significant improvements in results.

## 1 Introduction

Graphs can manifest the most rich form of data and are pervasively used in many real-world applications such as social networks (Tang et al., 2009; Chen et al., 2010; Qiu et al., 2018), citation networks (Gollapalli & Caragea, 2014), e-commerce (Baumann et al., 2018; Wang et al., 2019) and recommendation systems (Wu et al., 2022; Gao et al., 2023). The domain of Graph Representation Learning (GRL) has gained much traction in the last few years to model the rich information that graphs manifest and has established itself as a de-facto standard approach to solve several tasks such as graph classification, link prediction and node classification. A myriad of Graph Neural Network (GNN) algorithms have been proposed that mostly fall under the general notion of message passing, i.e., aggregation and updation (Kipf & Welling, 2017; Hamilton et al., 2017; Gilmer et al., 2017; Veličković et al., 2018; Xu et al., 2019; Abu-El-Haija et al., 2019; Pei et al., 2020; Yang et al., 2022). Message passing enables representation learning via iterative updation of a node's representation by aggregating the features of its neighbours. The selection of neighbours and the aggregation operation has led to the designing of a wide variety of algorithms over the last couple of years.

In general, real-world networks fall into either of the two categories, i.e. homophilic or heterophilic, decided by a network property called homophily. Homophily is the tendency to connect similar nodes via edge linkage, where class labels of the connected nodes generally govern the notion of similarity. For example, in citation networks, researchers often tend to cite research articles from the same domain (Ciotti et al., 2016). In contrast, low homophily, i.e., heterophily, is observed in heterophilic datasets, where edge formations do not favour similar class labels or actually favour dissimilar class labels. e.g. In social media platforms, people tend to form connections irrespective of gender, whereas, in dating networks, most people prefer to form connections with the opposite gender (Zhu et al., 2021).

A large number of GNN algorithms tend to perform better on homophilic graphs (Xu et al., 2018; Gasteiger et al., 2019; Wu et al., 2019; Deng et al., 2020; Bojchevski et al., 2020; Huang et al.,

2021; Brody et al., 2022) and are assumed to be not suitable for graphs with heterophily (Zhu et al., 2021; Wang et al., 2022; He et al., 2022). This assumption has led to the designing of specialised algorithms for heterophilic datasets. In the recent years, various algorithms have been proposed specifically for heterophilic datasets (Jin et al., 2021; Chen et al., 2020; Chien et al., 2021; Zhu et al., 2020; Lim et al., 2021; Bodnar et al., 2022; Li et al., 2022; Zheng et al., 2022). As highlighted by Platonov et al. (2023), these recently proposed heterophilic GNNs are evaluated on six heterophilic datasets used by Pei et al. (2020) wherein two datasets have a major drawback of train-test data leakage due to the presence of duplicate nodes. Recently Lim et al. (2021) released several new large-scale and diverse heterophilic datasets.

To this end, we propose Spectral Highways, a novel technique that enriches a given heterophilic graph dataset with additional nodes and connections forming highways over the original graph. These highways enable better information exchange between different spectral regions of the heterophilic graph, boosting the performance of both heterophilic and non-heterophilic GNNs for node classification. We relate the performance of Spectral Highways with Adjusted homophily across the heterophilic datasets. We also discuss the applicability of Spectral Highways for homophilic datasets in Section 5.

## 2 RELATED WORK

### 2.1 GRAPH DATASETS

**Homophilic datasets**   Preliminary research works in GRL mainly evaluated their algorithms on datasets that possess high homophily. The most widely used datasets for benchmarking are three citation networks, namely Citeseer, Cora and Pubmed (Giles et al., 1998; Sen et al., 2008; McCallum et al., 2000; Namata et al., 2012; Yang et al., 2016), and two co-purchasing networks, namely amazon-photo and amazon-computers (Shchur et al., 2018). Other homophilic datasets used for node classification are citation co-author networks: coauthor-cs and co-author-physics from (Shchur et al., 2018). To evaluate GNNs on large-scale datasets, Hu et al. (2020) created Open Graph Benchmark and introduced highly homophilic datasets for node classification: ogbn-products, ogbn-arxiv, ogbn-proteins, ogbn-mag and ogbn-papers100M.

**Heterophilic datasets**   Pei et al. (2020) introduced six graph datasets possessing high heterophily that prompted the designing of specific methods for heterophilic graphs. These six graphs, namely Squirrel, Chameleon, Actor, Texas, Wisconsin, and Cornell, have become the standard benchmarks for evaluating heterophilic GNNs. Platonov et al. (2023) corrected the node duplication in Squirrel and Chameleon datasets and introduced Squirrel filtered and Chamaleon filtered datasets along with five new medium-size datasets: roman-empire, amazon-ratings, minesweeper, tolokers, and questions. Lim et al. (2021) released seven new large-scale heterophilic datasets, namely Penn94, pokec, arXiv-year, snap-patents, genius, twitch-gamers, and wiki.

### 2.2 GRL ALGORITHMS

**Non-heterophilic GNNs**   GNNs have shown their effectiveness on a wide variety of graph learning tasks on real-world datasets. The majority of GNN algorithms are based on the convolution principle which is defined as neighbourhood aggregation and updation. GCN (Kipf & Welling, 2017) aggregates the features of a node's neighbours by learning a weight matrix and uses them to update the node's feature vector. GraphSAGE (Hamilton et al., 2017) samples nodes from the 1-hop and 2-hop neighbourhood for aggregation. GAT (Veličković et al., 2018) uses an attention mechanism to give varied importance to various neighbours. Xu et al. (2018) introduced Jumping Knowledge networks to capture varied neighbourhood ranges for different nodes where subgraphs have diverse local structures. Wu et al. (2019) proposed a Simple Graph Convolution by successively dropping non-linearities and collapsing weight matrices between consecutive network layers, resulting in a linear classifier following a low pass filter. Gasteiger et al. (2019) explored the relationship between personalised PageRank and GCN to fast approximate the propagation of neural predictions. Brody et al. (2022) designed GATv2 to introduce dynamic attention by reversing the order of attention and non-linearity operations in GAT.

**Heterophilic GNNs** Pei et al. (2020) directed focus towards heterophilic datasets by introducing Geom-GCN that does bi-level aggregation over the structural neighbourhood obtained by mapping the original graph into a latent continuous space. Zhu et al. (2020) discussed the limitations of GNNs for learning under heterophily and proposed H2GCN. Zhu et al. (2021) proposed CPGNN to learn a class compatibility matrix to model graph homophily. Chien et al. (2021) proposed the use of Generalised PageRank (GPR) for GNN where GPR weights automatically learn to adjust weights in accordance with node label pattern. Lim et al. (2021) proposed LINKX, a simple technique of embedding adjacency matrix and node features separately through MLPs and combining them by concatenation. Fu et al. (2022) introduced $p$-Laplacian based GNN as an approximation of a polynomial graph filter over the spectral domain of $p$-Laplacians. Wang et al. (2022) suggested an adaptive propagation mechanism and aggregation process as per the homophily between node pairs based on attribute and topological information. Li et al. (2022) suggested two models, GloGNN and GLoGNN++, that capture node correlations by learning a coefficient matrix to guide the neighbourhood aggregation further. Maurya et al. (2022) designed FSGNN highlighting the use of softmax as a regulariser and soft-selector of neighbourhood features. Bodnar et al. (2022) proposed neural sheaf diffusion models to achieve linear discrimination of classes in the infinite time limit. GBK_GNN Du et al. (2022) suggested the use of bi-kernel feature transformation to capture homophily and heterophily followed by a selection gate over kernels for given node pairs. He et al. (2022) suggested block-guided classified aggregation to learn separate aggregation rules for neighbours of varied classes. Cavallo et al. (2023) proposed incorporating a learnable importance coefficient per layer to balance the contributions of the neighbourhood and the ego node. Zheng et al. (2023) proposed neural architecture search to build heterophilic GNN models automatically.

## 3 PROPOSED TECHNIQUE

### 3.1 MOTIVATION

In this section, we discuss the motivation behind our proposed technique, Spectral Highways. The three primary factors that have driven the research for specialised GNNs for heterophilic graphs are (i) the assumption that most GNNs perform better on homophilic graphs, (ii) in heterophilic networks, vertices with high structural and semantic similarities are generally farther away from each other, and (iii) uniform neighbourhood aggregation and updation is oblivious to the information between similar and dissimilar neighbours. As discussed in Section 2.2, many specialised methods have attended to the above factors. Spectral Highways leverages and challenges the above assumption by injecting homophily into heterophilic graphs as shown empirically in Section 4 and Section 5. Spectral Highways enables information flow between different regions of the heterophilic graph and thus brings semantic and structural similar vertices close to each other. Spectral Highways is a technique designed from the data engineering perspective to enrich the existing graph data and thus allows available heterophilic and non-heterophilic GNNs to operate upon it.

### 3.2 SPECTRAL HIGHWAYS

Spectral Highways (as shown in Fig.1) is a network of highways that run over the top of regions formed by Spectral Clustering over a graph. Spectral Clustering uses connectivity information between data points to form clusters using eigenvalues and eigenvectors of the data matrix. Let $G = (V, E)$ be a undirected graph with vertex set $V = \{v_1, v_2, \ldots, v_n\}$ and edge set $E$. Let $W = (w_{ij})_{i,j=1,\ldots,n}$ be the weighted adjacency matrix of the graph $G$ where $w_{ij}$ represents the edge weight between nodes $v_i$ and $v_j$. If the graph is unweighted, then $w_{ij} = 1$ for an edge present between nodes $v_i$ and $v_j$; otherwise $w_{ij} = 0$. Let $d_i = \sum_{j=1}^{n} w_{ij}$ be the degree of a node $v_i \in V$ and we define degree matrix $D$ as a diagonal matrix with degrees $d_1, \ldots, d_n$ on its diagonal. Then, we can define the unnormalised graph Laplacian matrix as $L = D - W$.

We perform Spectral Clustering according to the procedure laid down by Shi & Malik (2000). Let $K$ be the number of clusters we want to construct in $G$. Then, we compute the first $K$ generalised eigenvectors $u_1, \ldots, u_n$ of the generalised eigenproblem $Lu = \lambda Du$. We then stack $u_1, \ldots, u_n$ as column vectors to construct $U \in \mathbb{R}^{n \times K}$. We do not use the popular k-means algorithm (Lloyd, 1982) as it is an iterative scheme sensitive to initialisation, which can lead to poor clusterings. We then directly extract clusters from eigenvectors by cluster_qr method (Damle et al., 2019).

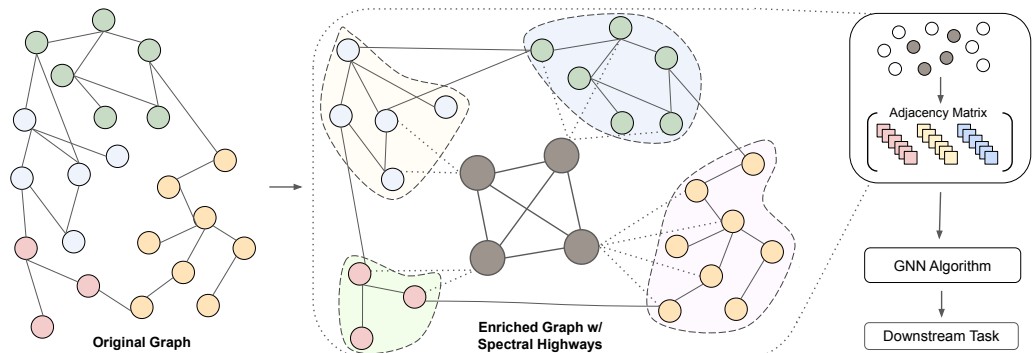

Figure 1: Overview of the use of Spectral Highways. For a given heterophilic graph, we use Spectral Highways to construct an enriched graph. We run available heterophilic or non-heterophilic GNN algorithm on the enriched graph for a downstream node classification task. In this representative enriched graph, the values of $K$, $mincon$ and $pcon$ are 4, 2 and 0.5 respectively.

Let $C = \{c_1, \ldots, c_K\}$ be the set of clusters obtained by Spectral Clustering where each such cluster represents a subgraph or a region formed corresponding to the graph topology. We construct highways over the obtained spectral clusters to allow information exchange between different regions of the graph. We instantiate a new node called Spectral node for $c_i \in C \ \forall \ i \in \{1, \ldots, K\}$. We then connect these Spectral nodes among each other to form a network layer. To construct highways, we need to connect the network of spectral nodes to the underlying graph.

For each Spectral node $s_i$, we connect it to the corresponding spectral cluster $c_K$ via a suitable connectivity principle. Instead of making random connections, we define the connectivity principle based on node importance. We propose the use of two popular algorithms to rank the node importance: PageRank and DivRank. PageRank determines a node's importance by considering the incoming edges it receives from other important nodes in the graph. It outputs a probability distribution over the network to represent the likelihood of a random surfer arriving at a particular node. We assume a uniform initial probability distribution at time step t=0 such that the PageRank score of a node $v_i$ is $R(i; 0) = 1/n$. Iteratively, at any time step $t$, PageRank score is

$$R(i; t + 1) = \frac{1 - d}{n} + d \sum_{j \in M(i)} \frac{R(j; t)}{L(j)} \tag{1}$$

where $M(i)$ represents the incoming neighbours of node $v_i$, $L(j)$ is the number of outgoing links of node $v_j$ and d is the damping factor. The value of d is generally taken as 0.85 and corresponds to the probability that a random surfer continues to follow the outgoing links at node $v_i$.

PageRank relates to the prestige of the nodes in a network, but diversity is another important property that we can account for ranking important nodes. DivRank ranks nodes in a network by setting up an interplay between prestige and diversity. DivRank ensures a wide coverage for ranking node importance, whereas PageRank can lead to rank top nodes in a 1-hop neighbourhood. Similar to PageRank, DivRank outputs a probability distribution over the network, indicating the node ranking.

We then connect a Spectral node $s_i$ to a certain number of nodes in the spectral cluster $c_i$ based on a percentage connectivity parameter $pcon$ consistent across all clusters. We choose percentage as the connectivity measure rather than a fixed integer because spectral clusters are of variable sizes. It ensures that we have a uniform extent of coverage across clusters. For small datasets, we can observe a few clusters that are small in size such that they end up having zero connections as per $pcon$. To account for this scenario, we introduce a $mincon$ parameter that ensures a minimum number of connections to be formed. Still, if the cluster size is too small to accommodate the $mincon$, we do not connect to that cluster and drop the corresponding spectral node.

We have discussed the ranking algorithms and the connectivity coverage above for our connectivity principle. These offer us two new hyperparameter choices, namely $mode$ and $ctype$. We choose $mode$ as a hyperparameter to decide whether to run ranking on a cluster level or graph level, i.e., `local` or `global`. $ctype$ decides the type of nodes to choose for making connections. We explore

four different ways to select from ranked nodes: `low`, `mid`, `high` and `lmh`. Opting `low` enables connections to the nodes at the bottom of the ranked node spectrum. Similarly, `mid` and `high` lead to connection formation to the nodes in the middle and at the top of the ranked node spectrum, respectively. `lmh` enforces an equally distributed number of link formations with each of the low, mid and high ranked nodes. Intuitively, it may appear to make connections only to the highly important nodes, but empirically, results show no absolute winner for the best choice of $ctype$. Similarly, for $mode$, it may sound better to focus on the local level than the global one, as the Spectral nodes are already connected in a separate network layer to account for global information exchange. However, exhaustive experimentation indicates not to favour any particular $mode$ type.

The above steps ensure the structural formation of Spectral Highways where nodes (not all) via a highway of Spectral nodes interact with other nodes (not all) in the farther regions in the graph as well as in the same spectral cluster, leading to an enhanced information flow. To initialise the embeddings of a Spectral node, we would not want to compute the average of the representations of nodes forming a connection with it, as this will lead to oversmoothing. Hence, we initialise the embedding of a spectral node with a random sequence of zeroes and ones keeping the same embedding dimension as those of its neighbouring nodes.

To assign a class label to each Spectral node, we take the majority voting of class labels of nodes belonging to the cluster and assign it as the class label of the Spectral node. This step leads to injecting homophily into the heterophilic graph. Furthermore, the previous step of initialising Spectral node embedding ensures that we inject homophily in a controlled fashion. To summarise, Spectral Highways offer the following hyperparameters to tune:

- Number of spectral clusters, $K$: $\{5, 10, 20, 30, 40, 50, 60, 65, 70\}$
- Choice of ranking algorithm, $rtype$: $\{\texttt{PageRank}, \texttt{DivRank}\}$
- Percentage connectivity, $pcon$: $\{0.1, 0.2, 0.3, 0.4, 0.5, 0.6\}$
- Minimum number of connections, $mincon$: $\{2, 3\}$
- Mode of ranking, $mode$: $\{\texttt{local}, \texttt{global}\}$
- Connectivity type, $ctype$: $\{\texttt{low}, \texttt{mid}, \texttt{high}, \texttt{lmh}\}$

Mathematically, we describe Spectral Highways (SH) for a given input graph $G(V, E)$ as a data engineering technique outlined by the following process:

$$SH(G(V, E)) \Rightarrow G'(V', E') \leftrightarrow G'(V + S, E + E'' + E''') \qquad (2)$$

where $S = \{s_1, \ldots, s_K\}$ is the set of Spectral nodes, $|E''| = \binom{K}{2}$ is cardinality of the set of connections formed amongst the Spectral nodes in the network layer and $E''' = \{N_e(s_1), \ldots, N_e(s_K)\}$.

$N_e(s_i)$ represents the edge neighbourhood of $s_i$ in the underlying graph $G$ and is given by

$$\begin{aligned} N_e(s_i) &= f(mincon, pcon, mode, ctype, rtype, C_i, G) \\ |N_e(s_i)| &= max(mincon, [pcon * |C_i|]_+) \end{aligned} \qquad (3)$$

where $[x]_+$ represents greatest integer less than or equal to $x$. Also, the embedding and the class label of Spectral node $s_i$ is as follows:

$$s_i = [rand\{0, 1\}]^d \; ; \; y(s_i) = M[y(c_i^1), y(c_i^2), \ldots, y(c_i^p)] \qquad (4)$$

where $d$ is the dimension of node features, $y$ is the class label, and $M$ is mathematical mode operator.

## 4 EXPERIMENTS

**Experimental setup**  We conduct extensive experimentation for node classification on a variety of heterophilic datasets and both heterophilic and non-heterophilic GNNs. As Spectral Highways augments the existing heterophilic graph, its merit is determined by the performance of downstream GNNs. We take a heterophilic graph and use Spectral Highways to generate an enriched graph and then run an available GNN model on this enriched graph to predict the class of a node. For a fair comparison, we only keep all the Spectral nodes in the train set and do not use them for validation or in the test set. Although it is not required, we still use the same hyperparameters of a GNN model

for Spectral Highways and the original given graph. Better results can be achieved using different GNN hyperparameters for Spectral Highways as the underlying graph is modified. For each dataset, we consider 5 different train/val/test splits and run 3 rounds of experiments for each of the splits. Following (Fu et al., 2022), we take 60/20/20 as the train/val/test split ratio. All the experiments are run for 100 epochs. We choose the commonly used accuracy as a metric and report its mean and standard deviation over the 15 runs. We run all experiments on an NVIDIA DGX A100 40GB GPU.

Table 1: Performance comparison of Spectral Highways w.r.t. various models on eight heterophilic datasets. We report the accuracy values for GNN models and Spectral Highways with $rtype$: {PageRank, DivRank}. ChameleonF and SquirrelF represents the filtered versions of Chameleon and Squirrel datasets. Twitch denotes Twitch-Gamers dataset and Arxiv denotes arXiv-year dataset. We highlight the global best result across GNNs for each dataset. Furthermore, we highlight significant improvements (fourth bin) over baselines for each dataset.

| | Cornell | Texas | Wisconsin | Actor | ChameleonF | SquirrelF | Twitch | Arxiv |
|---|---|---|---|---|---|---|---|---|
| MLP | 84.34 ± 5.86 | 77.00 ± 12.98 | 94.81 ± 5.16 | 43.51 ± 2.72 | 54.13 ± 5.05 | 34.46 ± 10.48 | 60.67 ± 0.90 | 39.48 ± 2.26 |
| SH Page | **88.48 ± 6.08** | **80.08 ± 11.03** | **96.73 ± 1.67** | 41.63 ± 3.56 | **56.28 ± 3.12** | **44.34 ± 9.02** | 60.10 ± 1.13 | 40.31 ± 3.15 |
| SH Div | **88.48 ± 6.08** | **80.08 ± 11.03** | **96.73 ± 1.67** | 41.63 ± 3.56 | **56.28 ± 3.12** | **44.34 ± 9.02** | 59.83 ± 1.99 | 40.31 ± 3.15 |
| GCN | 53.23 ± 14.81 | 60.75 ± 20.44 | 67.72 ± 5.93 | 33.43 ± 3.60 | 57.81 ± 3.23 | 44.37 ± 3.58 | 58.60 ± 2.10 | 49.46 ± 1.96 |
| SH Page | **59.29 ± 16.30** | **64.17 ± 8.71** | **71.48 ± 3.62** | 34.14 ± 2.37 | 58.33 ± 3.62 | 44.65 ± 3.65 | **60.55 ± 0.33** | 43.38 ± 1.69 |
| SH Div | **55.56 ± 11.56** | **64.42 ± 3.66** | **70.80 ± 4.60** | 34.08 ± 2.50 | 58.30 ± 2.36 | 44.66 ± 3.03 | **60.33 ± 0.54** | 43.24 ± 2.28 |
| SAGE | 86.67 ± 4.10 | 71.83 ± 7.97 | 89.01 ± 5.98 | 40.46 ± 2.26 | 56.94 ± 3.80 | 40.25 ± 8.54 | 60.26 ± 0.63 | 50.17 ± 0.60 |
| SH Page | 81.92 ± 7.50 | **80.92 ± 7.93** | **94.32 ± 2.47** | 39.08 ± 1.75 | **59.17 ± 3.07** | **42.20 ± 2.81** | 60.63 ± 0.37 | 43.78 ± 1.40 |
| SH Div | 81.31 ± 5.97 | **80.50 ± 11.40** | **94.14 ± 3.26** | 39.04 ± 1.80 | **59.65 ± 2.60** | **41.86 ± 3.79** | 60.39 ± 0.60 | 43.89 ± 1.83 |
| GAT | 45.96 ± 14.44 | 56.92 ± 20.98 | 64.94 ± 5.77 | 34.51 ± 1.80 | 58.51 ± 2.74 | 42.65 ± 7.26 | 52.04 ± 6.84 | 21.81 ± 4.07 |
| SH Page | **57.17 ± 11.52** | **64.50 ± 10.93** | **71.91 ± 4.37** | 33.70 ± 2.29 | **59.83 ± 3.78** | 43.58 ± 3.81 | **54.87 ± 4.88** | **45.79 ± 2.33** |
| SH Div | **56.36 ± 12.20** | **64.50 ± 11.06** | **72.59 ± 5.27** | 33.69 ± 2.40 | **59.72 ± 4.62** | 44.42 ± 4.58 | **53.91 ± 4.75** | **45.44 ± 2.52** |
| JKNet | 58.08 ± 15.95 | 59.08 ± 19.26 | 70.37 ± 5.17 | 37.91 ± 2.86 | 59.03 ± 1.61 | 47.53 ± 3.13 | 60.50 ± 0.89 | 47.50 ± 2.46 |
| SH Page | **61.11 ± 10.68** | **67.75 ± 6.93** | **73.70 ± 4.03** | **39.76 ± 1.54** | 59.03 ± 3.53 | **48.57 ± 3.46** | 60.42 ± 0.63 | 47.31 ± 2.33 |
| SH Div | **61.72 ± 9.57** | **65.33 ± 9.93** | **74.63 ± 3.93** | **39.79 ± 1.60** | 59.34 ± 3.76 | **48.60 ± 3.89** | 60.48 ± 0.53 | 47.26 ± 2.32 |
| SGC | 56.06 ± 9.60 | 66.83 ± 6.76 | 64.57 ± 6.38 | 32.83 ± 3.67 | 59.62 ± 3.45 | 48.17 ± 2.36 | 56.93 ± 0.78 | 49.21 ± 1.36 |
| SH Page | 55.96 ± 7.09 | 66.58 ± 7.70 | **70.06 ± 5.43** | **34.53 ± 3.32** | 58.30 ± 3.85 | **49.31 ± 1.75** | **58.79 ± 0.77** | 44.01 ± 1.81 |
| SH Div | **58.28 ± 5.28** | **70.33 ± 9.17** | **70.12 ± 4.00** | **34.71 ± 3.66** | 58.65 ± 3.40 | **49.47 ± 2.63** | **58.68 ± 0.88** | 43.32 ± 1.53 |
| APPNP | 86.06 ± 6.12 | 81.83 ± 5.09 | 96.60 ± 1.47 | 43.56 ± 3.70 | 59.93 ± 2.64 | 38.53 ± 4.18 | 60.66 ± 0.57 | 37.46 ± 6.24 |
| SH Page | 88.99 ± 3.99 | **85.42 ± 5.25** | 97.10 ± 1.81 | 42.22 ± 2.91 | 62.60 ± 2.31 | **41.27 ± 8.29** | 60.51 ± 0.66 | **44.56 ± 1.30** |
| SH Div | 88.99 ± 3.99 | **85.42 ± 5.25** | 97.10 ± 1.81 | 42.22 ± 2.91 | 62.12 ± 1.79 | **41.59 ± 9.43** | 60.51 ± 0.66 | **44.79 ± 1.10** |
| GPRGNN | 82.02 ± 9.93 | 75.75 ± 12.29 | 92.96 ± 3.18 | 41.80 ± 2.09 | 60.52 ± 2.94 | 45.91 ± 3.90 | 59.83 ± 1.15 | 21.58 ± 6.76 |
| SH Page | **84.44 ± 5.33** | **85.83 ± 4.60** | 91.85 ± 3.47 | 39.40 ± 3.56 | **61.60 ± 2.38** | **49.03 ± 5.29** | 59.45 ± 0.81 | **45.88 ± 1.13** |
| SH Div | **84.65 ± 4.79** | 86.33 ± 4.32 | 92.10 ± 3.32 | 39.47 ± 3.63 | **61.84 ± 2.58** | 49.63 ± 3.27 | 59.45 ± 0.81 | **46.15 ± 0.87** |
| LINKX | 67.88 ± 14.22 | 62.42 ± 14.60 | 81.17 ± 9.27 | 33.88 ± 3.55 | 57.74 ± 2.98 | 43.14 ± 8.33 | 62.62 ± 1.62 | 52.94 ± 2.43 |
| SH Page | **77.37 ± 12.43** | **76.67 ± 9.27** | **91.54 ± 4.82** | **36.20 ± 4.76** | **59.41 ± 4.95** | **48.43 ± 6.27** | 65.40 ± 1.54 | 45.68 ± 2.47 |
| SH Div | **77.98 ± 8.57** | **77.00 ± 11.65** | **91.11 ± 5.54** | **36.17 ± 3.76** | **59.10 ± 4.20** | **46.60 ± 6.16** | 65.14 ± 2.13 | 44.71 ± 4.43 |
| GATv2 | 39.49 ± 22.88 | 48.67 ± 28.00 | 65.06 ± 8.24 | 33.27 ± 1.87 | 57.60 ± 2.98 | 42.56 ± 6.15 | 50.60 ± 7.12 | 24.86 ± 7.94 |
| SH Page | **53.03 ± 14.59** | **66.58 ± 13.53** | **69.69 ± 4.41** | 32.13 ± 2.39 | **60.21 ± 2.60** | **44.58 ± 3.50** | **57.61 ± 2.32** | **46.09 ± 2.36** |
| SH Div | **56.06 ± 15.25** | **64.50 ± 18.30** | **70.12 ± 5.94** | 32.28 ± 3.05 | **60.03 ± 3.13** | **45.38 ± 2.31** | **57.05 ± 3.28** | **46.06 ± 2.29** |
| pGNN | 73.03 ± 10.41 | 68.83 ± 8.58 | 80.06 ± 6.87 | 33.79 ± 2.18 | 58.19 ± 3.84 | 48.90 ± 3.58 | 60.86 ± 1.08 | 41.11 ± 0.75 |
| SH Page | **79.19 ± 5.29** | **74.67 ± 9.26** | **86.91 ± 4.71** | **34.93 ± 2.09** | 58.96 ± 2.19 | 47.84 ± 5.79 | 60.89 ± 1.14 | **42.25 ± 1.40** |
| SH Div | **78.48 ± 4.83** | **76.67 ± 9.30** | **87.16 ± 2.34** | **34.94 ± 1.81** | 58.61 ± 3.44 | 47.76 ± 4.45 | 60.71 ± 1.29 | **42.32 ± 1.54** |

**Baseline GNNs** We employ various neural architectures as baseline models and compare their respective performances with the use of Spectral Highways. We choose **MLP** (Goodfellow et al., 2016) as our first baseline as it is a simple neural architecture that considers only node features and does not consider the graph topology. We use **GCN** (Kipf & Welling, 2017) that fuelled the research in GRL by introducing convolution in graphs. We take **GraphSAGE** (Hamilton et al., 2017), which is another classic GNN architecture. We then choose **GAT** (Veličković et al., 2018), which uses an attention mechanism for neighbourhood aggregation. We also use **JKNet** (Xu et al., 2018) that flexibly leverage different neighbourhood ranges for better structure-aware learning. We choose one simple method **SGC** (Wu et al., 2019) that learns a linear classifier following a low pass filter by successively dropping non-linearities and collapsing weight matrices between consecutive layers. We take **APPNP** (Gasteiger et al., 2019) as our next baseline as it explores the relationship between personalised PageRank and GCN to fast approximate the propagation of neural predictions. We then choose **GPRGNN** (Chien et al., 2021) that uses generalised PageRank for GNN. We use **LINKX** (Lim et al., 2021) that learns to embed adjacency matrix and node features separately through MLPs. We use **GATv2** (Brody et al., 2022), which introduces dynamic attention by reversing the order of attention and non-linearity operations in GAT. At last, we take **pGNN** (Fu et al.,

2022) that introduces $p$-Laplacian based GNN as an approximation of a polynomial graph filter over the spectral domain of $p$-Laplacians. So, we choose a wide variety of GNNs for exhaustive baseline comparisons: **Only node features** (MLP), **Simple method** (SGC), **Homophilic GNNs** (GCN, GraphSAGE, GAT, APPNP, JKNet, GATv2) and **Heterophilic GNNs** (GPRGNN, LINKX, pGNN).

**Benchmark datasets** For benchmarking and evaluating the performance of our proposed technique, we choose eight datasets with varied statistics, as shown in Table 5. We choose **Cornell**, **Texas**, **Wisconsin**, and **Chameleon filtered** heterophilic datasets for their small size, **Squirrel filtered** and **Actor** datasets for their medium size, and **Twitch-Gamers** and **arXiv-Year** datasets for their large size. We could not take other datasets like pokec, genius, wiki, etc., as their experiments ran out of memory. Cornell, Texas and Wisconsin are datasets of WebKB [1] page data gathered from computer science departments of various universities. Squirrel and Chameleon datasets are introduced for node prediction by Pei et al. (2020) and have been extensively used for evaluating heterophilic GNNs. Recently, Platonov et al. (2023) identified the issue of node duplication in these datasets and released their corrected versions, namely Squirrel filtered and Chamaleon filtered. Lim et al. (2021) introduced Twitch-Gamers and arXiv-Year datasets with the task of predicting the presence of mature content in user accounts of online social networks, and year of publication or patent grant in citation network, respectively.

**Results** Table 1 shows the performance of several models with and without the incorporation of Spectral Highways on various heterophilic datasets. We present the results of Spectral Highways (SH) for both variants of the ranking algorithm, PageRank and DivRank. We have 11 baseline GNNs and 8 heterophilic datasets, resulting in 88 benchmark combinations. We segregate the results into four bins to evaluate the performance of our technique. The first bin corresponds to the results where SH Page/SH Div perform poor than the baseline model. In the second bin, we consider results where we lag behind the baseline results by 1 unit. The third bin corresponds to the results where we perform better than baseline but not more than 1 unit. At last, we consider the cases where we outperform the baselines significantly. From the results table, we can see that the bin distribution is $10, 8, 10, 60$ respectively. We can see that Spectral Highways significantly outperform the baselines on 60 benchmark combinations giving an average of 14% increase in accuracy. We can observe that both variants of Spectral Highways give similar results. Also, we achieve the best performance across all models on six out of eight datasets.

## 5 ANALYSIS AND DISCUSSION

As discussed in Section 4, Spectral Highways gives superior performance on several heterophilic datasets and downstream GNN models. We design Spectral Highways to bring close structurally and semantically similar nodes that are generally present in faraway regions in a heterophilic graph. To evaluate the assumption that most GNNs perform better on graphs with high homophily, we explored several homophily measures that are available in the literature. Let $G = (V, E)$ is a graph with $n$ nodes, and each node $u \in V$ has a class label $y_u \in \{0, 1, \ldots, C - 1\}$, where $C$ is the total number of classes and $C_k$ represents the set of nodes in class $k$. Node homophily (Pei et al., 2020), which computes the ratio of neighbours that have the same class for an ego node and then computes the mean of these ratios across all nodes, is defined as

$$H_{node} = \frac{1}{|V|} \sum_{v \in V} \frac{|\{u \in N(v) : y_v = y_u\}|}{|N(v)|} \tag{5}$$

where $N(v)$ denotes the neighbourhood of node $v$ Edge homophily (Zhu et al., 2020) is another standard measure for homophily, which is the fraction of edges connecting two nodes with the same class:

$$H_{edge} = \frac{|\{(v, u) \in E : y_v = y_u\}|}{|E|} \tag{6}$$

Lim et al. (2021) showed that these two simple and intuitive homophily measures are sensitive to the number of classes and their balance, and proposed an Improved homophily measure defined as

$$H_{imp} = \frac{1}{C - 1} \sum_{k=0}^{C-1} [h_k - \frac{|C_k|}{n}]_+ \tag{7}$$

---

[1] http://www.cs.cmu.edu/afs/cs.cmu.edu/project/theo-11/www/wwkb/

where $[a]_+ = max(a, 0)$, and $h_k$ is the class-wise homophily metric defined as

$$h_k = \frac{\sum_{u \in C_k} |\{u \in N(v) : y_v = y_u|}{\sum_{u \in C_k} |N(v)|} \tag{8}$$

Platonov et al. (2022) showed that Improved homophily can also lead to unreliable results and thus proposed a new measure, Adjusted homophily, by correcting the number of intra-class edges by their expected value and is thus insensitive to the number of classes and their balance. Adjusted homophily is based on Edge homophily and is computed as

$$H_{adj} = \frac{H_{edge} - \sum_{k=1}^{C} D_k^2/(2|E|)^2}{1 - \sum_{k=1}^{C} D_k^2/(2|E|)^2} \tag{9}$$

where $D_k = \sum_{v:y_v=k} d(v)$ and $d(v)$ represents the degree of a node $v$.

Table 2: Analysis of homophily scores for different homophily measures across eight heterophilic datasets. It shows the injection of homophily into heterophilic datasets using Spectral Highways.

| | Cornell | Texas | Wisconsin | Actor | ChameleonF | SquirrelF | Twitch-Gamers | arXiv-Year |
|---|---|---|---|---|---|---|---|---|
| Adjusted Homophily | | | | | | | | |
| Original graph | -0.2029 | -0.226 | -0.1323 | 0.0061 | 0.0347 | 0.0115 | 0.0899 | 0.0051 |
| Spectral Highways | -0.0771 | -0.045 | -0.0041 | 0.0075 | 0.0441 | 0.0175 | 0.0903 | 0.0179 |
| Node Homophily | | | | | | | | |
| Original graph | 0.1182 | 0.0872 | 0.1706 | 0.2219 | 0.2481 | 0.1961 | 0.5563 | 0.2893 |
| Spectral Highways | 0.2211 | 0.2301 | 0.2831 | 0.2282 | 0.2587 | 0.2152 | 0.5588 | 0.2852 |
| Edge Homophily | | | | | | | | |
| Original graph | 0.1321 | 0.1118 | 0.2061 | 0.2194 | 0.2403 | 0.2095 | 0.5452 | 0.2181 |
| Spectral Highways | 0.2633 | 0.3029 | 0.3434 | 0.2275 | 0.2459 | 0.2146 | 0.5453 | 0.2253 |
| Improved Homophily | | | | | | | | |
| Original graph | 0.0499 | 0 | 0.0495 | 0.0074 | 0.0465 | 0.0409 | 0.0899 | 0.0671 |
| Spectral Highways | 0.0429 | 0.0161 | 0.0525 | 0.0154 | 0.0461 | 0.0406 | 0.0902 | 0.0644 |

Table 3: Analysis of homophily scores for different homophily measures across five homophilic datasets. It shows the decrease in homophily for homophilic datasets using Spectral Highways.

| | Cora | Citeseer | Photo | Computers | Pubmed |
|---|---|---|---|---|---|
| Adjusted Homophily | | | | | |
| Original graph | 0.7712 | 0.6706 | 0.7851 | 0.6823 | 0.6861 |
| Spectral Highways | 0.6116 | 0.4701 | 0.7573 | 0.6622 | 0.5516 |
| Node Homophily | | | | | |
| Original graph | 0.8251 | 0.7062 | 0.8365 | 0.7853 | 0.7924 |
| Spectral Highways | 0.7582 | 0.6261 | 0.8062 | 0.7723 | 0.7213 |
| Edge Homophily | | | | | |
| Original graph | 0.8099 | 0.7355 | 0.8272 | 0.7772 | 0.8023 |
| Spectral Highways | 0.6854 | 0.5776 | 0.8062 | 0.7651 | 0.7209 |
| Improved Homophily | | | | | |
| Original graph | 0.7657 | 0.6267 | 0.7722 | 0.7001 | 0.6641 |
| Spectral Highways | 0.6271 | 0.4471 | 0.7508 | 0.6837 | 0.5379 |

We compute all the above-discussed homophily scores on the given heterophilic datasets before and after using Spectral Highways. From Table 2, we can observe that Spectral Highways consistently increases the Adjusted homophily scores across all the datasets. We observe similar trends for Node homophily and Edge homophily measures. As shown in Platonov et al. (2022), Improved homophily leads to unreliable results with no clear pattern in the scores. Analysing the results from Table 1 and the homophily scores, we can observe that Spectral Highways achieve better results in datasets where it leads to high increase in homophily scores.

To further verify our hypothesis empirically, we conduct another set of experiments on five commonly used homophilic datasets, namely Cora, Citeseer, Photo, Computers, and Pubmed. We show

the statistics of these five datasets in Table 6. We performed a similar experimental setup for homophilic datasets as used for heterophilic datasets. The node prediction results are shown in Table 4 and the homophily scores are reported in Table 3. We observe that using Spectral Highways for homophilic graphs leads to a decrease in the homophily level as measured by all four available homophily measures. The effect of homophily reduction clearly reflects the drop in performance across almost every homophilic dataset and the chosen GNN. We see a slight improvement in MLP performance as it does not consider the graph topology. These observations are further in sync with the trend observed for the Twitch-Gamers dataset, where it observes the least increase in homophily score as it already possesses comparatively high initial homophily.

The exhaustive experimentation provides ample empirical evidence that homophily is a desired network property, enabling most GNNs to perform better. We show empirically that Spectral Highways injects homophily in the heterophilic datasets, leading to better prediction performance for the downstream GNNs. Thus, we both challenge and leverage the common assumption that most GNNs perform better on high homophily datasets by injecting homophily into heterophilic datasets.

Table 4: Performance comparison of Spectral Highways w.r.t. various models on five homophilic datasets. We report accuracy for GNN models and Spectral Highways. Performance drop is observed for all GNNs as expected due to decrease in homophily. We observe few outliers on Pubmed.

|  | Cora | Citeseer | Photo | Computers | Pubmed |
|---|---|---|---|---|---|
| MLP | 76.64 ± 1.37 | 77.03 ± 1.23 | 90.44 ± 2.69 | 86.47 ± 1.58 | 88.69 ± 0.57 |
| SH Page | 77.14 ± 1.49 | 77.11 ± 1.16 | 91.44 ± 1.34 | 86.82 ± 1.35 | 88.68 ± 0.42 |
| SH Div | 77.14 ± 1.49 | 77.11 ± 1.16 | 91.44 ± 1.34 | 86.82 ± 1.36 | 88.68 ± 0.42 |
| GCN | 88.50 ± 0.92 | 81.76 ± 1.27 | 91.57 ± 6.00 | 86.76 ± 3.20 | 89.56 ± 1.04 |
| SH Page | 85.17 ± 0.92 | 80.41 ± 1.40 | 79.59 ± 16.90 | 83.47 ± 4.72 | 88.57 ± 0.33 |
| SH Div | 85.08 ± 1.04 | 80.37 ± 1.67 | 79.80 ± 15.75 | 83.16 ± 3.81 | 88.56 ± 0.40 |
| GraphSAGE | 88.93 ± 1.22 | 81.44 ± 1.49 | 92.41 ± 1.91 | 87.10 ± 1.60 | 90.83 ± 0.47 |
| SH Page | 85.04 ± 1.63 | 78.05 ± 1.51 | 91.80 ± 2.04 | 86.52 ± 1.28 | 90.33 ± 0.47 |
| SH Div | 84.91 ± 1.69 | 78.05 ± 1.53 | 91.79 ± 2.09 | 86.52 ± 1.22 | 90.35 ± 0.44 |
| GAT | 89.42 ± 0.73 | 82.12 ± 1.46 | 93.85 ± 0.59 | 89.29 ± 0.85 | 90.31 ± 0.30 |
| SH Page | 85.77 ± 0.73 | 80.98 ± 1.28 | 92.51 ± 0.53 | 87.58 ± 0.67 | 88.78 ± 0.26 |
| SH Div | 85.88 ± 1.08 | 80.94 ± 1.37 | 92.57 ± 0.54 | 87.66 ± 0.81 | 88.79 ± 0.36 |
| JKNet | 88.94 ± 1.10 | 81.92 ± 1.49 | 93.57 ± 0.84 | 89.34 ± 1.05 | 91.30 ± 0.33 |
| SH Page | 85.44 ± 0.97 | 80.75 ± 1.27 | 91.62 ± 1.46 | 88.19 ± 0.90 | 89.46 ± 0.41 |
| SH Div | 85.58 ± 1.00 | 80.85 ± 1.24 | 91.49 ± 1.66 | 88.19 ± 0.94 | 89.47 ± 0.38 |
| SGC | 89.29 ± 1.29 | 81.52 ± 1.39 | 93.61 ± 0.75 | 88.95 ± 0.50 | 86.98 ± 0.55 |
| SH Page | 85.69 ± 2.23 | 79.87 ± 1.06 | 92.53 ± 0.65 | 87.39 ± 0.34 | 87.85 ± 0.27 |
| SH Div | 85.95 ± 1.62 | 79.91 ± 1.33 | 92.50 ± 0.65 | 87.42 ± 0.48 | 87.86 ± 0.25 |
| APPNP | 88.83 ± 0.65 | 81.73 ± 1.73 | 94.48 ± 0.85 | 88.73 ± 0.77 | 89.25 ± 0.48 |
| SH Page | 84.86 ± 1.49 | 80.38 ± 1.26 | 94.14 ± 0.93 | 88.93 ± 0.98 | 89.51 ± 0.41 |
| SH Div | 84.96 ± 1.49 | 80.32 ± 1.39 | 94.09 ± 0.77 | 88.99 ± 0.94 | 89.52 ± 0.42 |
| GPRGNN | 89.76 ± 1.00 | 82.48 ± 1.73 | 93.26 ± 1.34 | 88.94 ± 1.18 | 91.56 ± 0.36 |
| SH Page | 86.61 ± 1.53 | 80.71 ± 1.31 | 92.06 ± 1.00 | 87.55 ± 1.07 | 89.82 ± 0.34 |
| SH Div | 86.58 ± 1.67 | 80.71 ± 1.22 | 92.00 ± 0.95 | 87.67 ± 1.03 | 89.80 ± 0.35 |
| LINKX | 81.22 ± 2.78 | 74.18 ± 1.27 | 94.65 ± 1.07 | 89.50 ± 1.03 | 88.09 ± 0.96 |
| SH Page | 76.22 ± 2.52 | 71.75 ± 1.85 | 93.83 ± 1.59 | 88.05 ± 1.00 | 87.81 ± 0.74 |
| SH Div | 75.85 ± 4.28 | 71.60 ± 2.41 | 93.65 ± 1.32 | 87.99 ± 0.96 | 87.83 ± 0.57 |
| GATv2 | 88.95 ± 1.05 | 82.06 ± 0.94 | 93.90 ± 0.80 | 90.19 ± 0.59 | 90.34 ± 0.33 |
| SH Page | 85.96 ± 1.23 | 80.91 ± 1.33 | 92.56 ± 0.55 | 88.38 ± 0.81 | 88.87 ± 0.38 |
| SH Div | 85.94 ± 1.28 | 80.79 ± 1.54 | 92.64 ± 0.50 | 88.41 ± 0.87 | 88.84 ± 0.30 |
| pGNN | 89.94 ± 1.43 | 81.28 ± 1.10 | 94.09 ± 0.91 | 89.30 ± 0.71 | 91.75 ± 0.29 |
| SH Page | 83.46 ± 1.42 | 78.48 ± 1.13 | 90.93 ± 1.32 | 85.09 ± 1.95 | 90.14 ± 0.82 |
| SH Div | 83.49 ± 1.12 | 78.48 ± 1.13 | 90.79 ± 1.25 | 84.92 ± 1.79 | 90.08 ± 0.78 |

## 6 CONCLUSION

In this paper, we introduce a perspective of data enrichment that enables better performance of heterophilic and non-heterophilic GNN algorithms on heterophilic graphs by injecting homophily. We propose Spectral Highways that enables better information flow between structurally and semantically similar nodes that may be present in faraway regions in the graph. We prove the effectiveness of our technique through several experiments and analyses. Our work highlights the importance of data enrichment rather than the need to design specialised models. In the future, we intend to utilise other network properties to learn richer representation for graphs.

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

## A   APPENDIX

Table 5: Statistics of chosen heterophilic datasets.

| Type | Dataset | # Nodes | # Edges | # Features | # Classes |
|---|---|---|---|---|---|
| WebKB Webpage | Cornell | 183 | 295 | 1703 | 5 |
| | Texas | 183 | 309 | 1703 | 5 |
| | Wisconsin | 251 | 499 | 1703 | 5 |
| Actor Co-occurrence | Actor | 7,600 | 33,544 | 931 | 5 |
| Wikipedia Webpage | Chameleon filtered | 890 | 8,904 | 2,325 | 5 |
| | Squirrel filtered | 2223 | 47,138 | 2,089 | 5 |
| Online Social Network | Twitch-Gamers | 168,114 | 6,797,557 | 7 | 2 |
| Citation Network | arXiv-Year | 169,343 | 1,166,243 | 128 | 5 |

Table 6: Statistics of chosen homophilic datasets.

| Dataset | # Nodes | # Edges | # Features | # Classes |
|---|---|---|---|---|
| Cora | 2,708 | 5,278 | 1,433 | 7 |
| Citeseer | 3,327 | 4,552 | 3,703 | 6 |
| Photo | 7,487 | 119,043 | 745 | 8 |
| Computers | 13,381 | 245,778 | 767 | 10 |
| Pubmed | 19,717 | 44,324 | 500 | 3 |

We use the official code repositories of the authors for implementing GPRGNN [2], pGNN [3] and LINKX [4]. For the rest of the baseline GNNs, we use the respective implementations in PyTorch Geometric provided by pGNN. We utilise scikit-learn (Pedregosa et al., 2011) implementation of Spectral Clustering.

---

[2]https://github.com/jianhao2016/GPRGNN

[3]https://github.com/guoji-fu/pGNNs

[4]https://github.com/CUAI/Non-Homophily-Large-Scale

