# OpenReview forum: "Spectral Highways: Injecting Homophily into Heterophilic Graphs"
_ICLR.cc/2024/Conference — ICLR 2024 Conference Withdrawn Submission_

### Official Review · Reviewer_okcj · 2023-10-22

**Soundness:** 2 fair
**Presentation:** 4 excellent
**Contribution:** 2 fair
**Rating:** 3
**Confidence:** 5

**Summary:**

This paper studies the node classification problem on graphs with agnostic homophily. Their proposed method is data engineering/augmentation-based, which inserts some super nodes into the given graph to improve the message passing among subgraphs. The experiments show that with the proposed spectral highway, many GNNs' performance is improved on various heterophilic graphs.

**Strengths:**

S1. The proposed method is data-centric, which aims to benefit many GNN models.

S2. The presentation of this paper is good and I can follow most of the ideas.

S3. The proposed method is concise and I think it is a promising solution to generalize GNNs by adding super nodes.

**Weaknesses:**

W1. The proposed method is highly heuristic-based and the fundamental reason why the proposed method can work is not clear.

W2. The experimental results are not convincing and significant enough.

W3. A minor weakness: similar ideas are studied in existing works, but not in the way of inserting super nodes.

I will elaborate more in details in the Questions section.

**Questions:**

Q1. One of the assumptions of this paper is that **"most GNNs perform better on homophilic graphs"**. I personally do not agree with it. A part of the heterophilic graphs are easy to fit, e.g., Wisconsin with 90+% accuracy, and some homophilic graphs are challenging. The difficulties of node classification on different datasets are not only related to the graph (label) homophily, but also related to the node features, and many other factors.

Q2. Overall the proposed method is heuristic-based. Again, I think inserting super nodes is a promising strategy, but it is not clear why adding super nodes according to the spectral clustering results can work. Also, at the top of Page 5, the paper claims **"results show no absolute winner for the best choice of ctype"** and **"exhaustive experimentation indicates not to favour any particular mode type"**, which is frustrating. It shows that the proposed method still requires many trials and errors without clear theoretical guidance.

Q3. I personally suggest moving Table 4 next to Table 1. **Experimental results on homophilic graphs and heterophilic graphs are equally important"** because in real-world applications we have no idea what is the (label) homophily of the given graph.

Q4. The performance improvement on the heterophilic graphs is just ok, not significantly. About 2/3 of cases show performance improvement. However, **the performance on the homophilic datasets shows that the proposed method can even degrade the GNN performance**, which is not expected and a main drawback of the proposed solution.

Q5. Some data engineering/augmentation-based solutions [1-3] are missing. They might need to be compared with the proposed method, or at least mentioned in the related work section. Specifically, **the idea of work [1] shares great similarities with the idea of this paper.**

[1] Suresh, Susheel, Vinith Budde, Jennifer Neville, Pan Li, and Jianzhu Ma. "Breaking the limit of graph neural networks by improving the assortativity of graphs with local mixing patterns." In Proceedings of the 27th ACM SIGKDD Conference on Knowledge Discovery & Data Mining, pp. 1541-1551. 2021.

[2] Xu, Zhe, Yuzhong Chen, Qinghai Zhou, Yuhang Wu, Menghai Pan, Hao Yang, and Hanghang Tong. "Node Classification Beyond Homophily: Towards a General Solution." In Proceedings of the 29th ACM SIGKDD Conference on Knowledge Discovery and Data Mining, pp. 2862-2873. 2023.

[3] Liu, Haoyu, Ningyi Liao, and Siqiang Luo. "SIMGA: A Simple and Effective Heterophilous Graph Neural Network with Efficient Global Aggregation." arXiv preprint arXiv:2305.09958 (2023).

---

### Official Review · Reviewer_r3Ms · 2023-10-26

**Soundness:** 2 fair
**Presentation:** 3 good
**Contribution:** 2 fair
**Rating:** 5
**Confidence:** 4

**Summary:**

The paper introduces a technique called Spectral Highways that aims to improve the performance of Graph Neural Network (GNN) algorithms on heterophilic datasets. The authors challenge the assumption that GNNs perform better on graphs with high homophily and propose Spectral Highways as a method to inject homophily into heterophilic graphs. This technique enhances connectivity and information flow between nodes in heterophilic graphs, leading to improved performance of both heterophilic and non-heterophilic GNN algorithms for node classification.

**Strengths:**

The authors did lots of experiments.

**Weaknesses:**

See below

**Questions:**

1. "in heterophilic networks, vertices with high structural and semantic similarities are generally farther away from each other" Any evidence for this claim?

2. Researchers already find that heterophily is not always harmful and homophily assumption is not always necessary for GNNs [2,3,4,5]. How does this paper align with these works?

3. How do you select spectral nodes and why? Why do you want to connect the spectral nodes?

4. What is the relation between better connectivity and addressing heterophily? The heterophily problem does not result from connectivity problem.

5. Missing related work and result comparison [1,2].

6. Comparison with other graph rewiring methods [6]. What is your advantage?


[1] Bernnet: Learning arbitrary graph spectral filters via bernstein approximation. Advances in Neural Information Processing Systems, 34, 14239-14251.

[2] Revisiting heterophily for graph neural networks. Advances in neural information processing systems, 35, 1362-1375.

[3] When do graph neural networks help with node classification: Investigating the homophily principle on node distinguishability. arXiv preprint arXiv:2304.14274.

[4] Demystifying Structural Disparity in Graph Neural Networks: Can One Size Fit All?. arXiv preprint arXiv:2306.01323.

[5] Is Homophily a Necessity for Graph Neural Networks?. In International Conference on Learning Representations 2022.

[6] Understanding over-squashing and bottlenecks on graphs via curvature. In International Conference on Learning Representations. 2022

---

### Official Review · Reviewer_eD98 · 2023-10-30

**Soundness:** 3 good
**Presentation:** 2 fair
**Contribution:** 2 fair
**Rating:** 5
**Confidence:** 4

**Summary:**

The paper introduces a graph construction technique, specifically a data augmentation method, to address the heterophily problem prevalent in graph datasets. The proposed approach, name as the Spectral Highways method, injects homophily into heterophilic graphs by reshaping the relationships between nodes within a neighborhood. This is achieved by reconstructing the adjacency matrices using spectral frequencies derived from nodes and structures.

The core concept of constructing spectral highways, incorporating spectral nodes, enables nodes along these highways to interact not only with nodes in their immediate neighborhood but also with nodes in more distant regions of the graph, provided they belong to the same spectral cluster. This data augmentation strategy has the potential to be seamlessly integrated into various graph neural network architectures, facilitating a smooth "homophilic message-passing" mechanism among nodes, thereby enhancing the accuracy of node classification tasks.

**Strengths:**

1. The method injects homophily into heterophilic graphs, as evident in Table 4, where the heterophilic nature of the graphs is reduced through the implementation of reconstructed spectral highways.

2. Spectral highways have the potential to enhance the performance of various graph neural network models when applied to heterophilic graph datasets, in the context of node classification tasks.

3. The paper's presentation and explanation of the proposed approach make it easily comprehensible and accessible for readers.

**Weaknesses:**

1. The discussion provided in the paper is not entirely convincing. The referenced work [1] demonstrates that the homophily assumption is not an absolute requirement for Graph Neural Networks (GNNs) to achieve strong performance in node classification tasks. Still, I agree that GNNs are good if homophily assumpotion holds. It is important to note that specific conditions can enable GNNs to perform reasonably well even in heterophilic graph datasets when those conditions are met, as discussed in [1].

2. The method clusters nodes based on spectral frequencies derived from the graph structure, computed using the unnormalized graph Laplacian (L) and the degree matrix (D). The reconstructed graph relies solely on the original graph structure and doesn't take into account node features or labels. [2] show that the homophily is closely related to either node features or node labels. This approach raises concerns regarding its effectiveness in reducing the heterophilic nature of the graphs. It is not entirely convincing that the reconstructed graphs will exhibit reduced heterophily when only the graph's structural information is considered for the reconstruction process. The paper should address this limitation and provide a more comprehensive explanation or potentially explore ways to incorporate node features or labels into the reconstruction process to potentially enhance its effectiveness.

3. It is stated that spectral highways can transition a graph from being heterophilic to more homophilic. In principle, one would anticipate that this technique should also be applicable to homophilic graph datasets, potentially enhancing their homophily further. This expectation is based on the idea that spectral highways, by design, should strengthen the homophily within a clustered neighborhood due to the similar spectral frequencies. However, Table 4 reveals a drop in the homophily level. This outcome is somewhat unexpected, as spectral highways were anticipated to be a more generalized technique. I recommend that the authors delve deeper into this unexpected result and provide a more detailed analysis. It's crucial to understand why spectral highways do not necessarily enhance the performance in homophilic graph datasets, especially when their principles seem to align with reinforcing homophily. A deeper insight into this aspect would greatly contribute to a better understanding of the spectral highways method and its applicability across a broader spectrum of graph datasets.

[1] When do graph neural networks help with node classification: Investigating the homophily principle on node distinguishability. Neurips 2023

[2] Revisiting Heterophily For Graph Neural Networks. Neurips 2022

**Questions:**

1. I suggest that the authors conduct a more thorough exploration of how their proposed spectral highways method aligns with the conditions outlined in [1]. It is essential for the paper to investigate whether spectral highways hold the potential to bolster these conditions, potentially resulting in an enhanced performance of Graph Neural Networks (GNNs) on heterophilic graph datasets. This in-depth analysis would greatly augment our understanding of the method's relevance and utility across a wider array of scenarios.

2. To assign labels to each Spectral node, the authors employ a majority voting approach based on the labels of nodes within the same cluster and then assign the resulting label. It raises a significant question about whether the labels of testing and validation nodes are similarly revealed to the spectral nodes. To gain a more comprehensive understanding, it would be valuable to see experiments where random labels are assigned to these spectral nodes. This exploration could shed light on how Graph Neural Networks (GNNs) perform in scenarios where spectral nodes have no knowledge of the true labels.

3. [2] introduces more precise homophily metrics that consider node-level features, labels, and graph structures. It would be insightful to observe how the implementation of spectral highways enhances these homophily measurements. This analysis could provide valuable insights into the impact of spectral highways on homophily levels.


[1] When do graph neural networks help with node classification: Investigating the homophily principle on node distinguishability. Neurips 2023

[2] Revisiting Heterophily For Graph Neural Networks. Neurips 2022

---

### Official Review · Reviewer_AHfa · 2023-11-02

**Soundness:** 2 fair
**Presentation:** 2 fair
**Contribution:** 2 fair
**Rating:** 5
**Confidence:** 5

**Summary:**

This paper introduces a method known as spectral highway for reconstructing graphs to improve performance when dealing with homophilic or heterophilic graphs. The method designs edges within and between clusters based on spectral clustering results and a set of hyper-parameters. The observed performance improvements underscore the method's effectiveness.

**Strengths:**

1. The designed graph construction methods is interesting and easy to follow.
2. The extensive experiments have been conduct in this paper, and the diverse datasets are incorporated.

**Weaknesses:**

1. This paper lack the comparisons with the literature methods which updating the graph structures.
When facing the heterophilic graphs, re-constructing the graph structures towards hemophiliac ones is a common strategy, and existing methods have wide explored on this[1]. This paper also conducted from this perspective, and it would be better to provide the comparisons on SH design and experiments.

2. The experimental evaluations on the designed method are insufficient.
	- On the design dimensions. The experiments are conducted with different aggregation operations and datasets. When refer to the design hyper parameters, e.g., clustering number, ranking algorithms are not mentioned, and these parameters used in Table 1 and 4 are not provided as well, which is insufficient when evaluating the effectiveness of the designed method.
	- Baseline selection. Besides, the baselines used in the experiments are kind of outdated. The papers published on ICML/KDD/ICLR 2023 can be considered.
	- The efficiency. This paper adopts the spectral cluster operation, while it is cost to clustering nodes or calculating the eigenvalues. It would be better to evaluate the efficiency of SH.


3. The organization of this paper can be improved.
	- The introduction of four evaluation metrics can be moved into Sec 2.
	- More reference can be added in Sec 3.

[1] Graph Neural Networks for Graphs with Heterophily: A Survey.

**Questions:**

How to understand "semantic similarities" in this paper. It seems that the new edges are selected based on the structures merely, from cluster design, ranking nodes and re-connections. How to understand  "semantic" in SH.